# Vitamin D status in breast cancer cases following chemotherapy: A pre and post observational study in a tertiary hospital in Yogyakarta, Indonesia

**Herindita Puspitaningtyas**[1], **Dian Caturini Sulistyoningrum**[2], **Riani Witaningrum**[3], **Irianiwati Widodo**[4], **Mardiah Suci Hardianti**[5], **Kartika Widayati Taroeno-Hariadi**[5], **Johan Kurnianda**[5], **Ibnu Purwanto**[5], **Susanna Hilda Hutajulu**[5]*

1 Master of Clinical Medicine Postgraduate Program, Faculty of Medicine, Public Health and Nursing, Universitas Gadjah Mada, Yogyakarta, Indonesia, 2 Department of Nutrition and Health, Faculty of Medicine, Public Health and Nursing, Universitas Gadjah Mada, Yogyakarta, Indonesia, 3 Division of Hematology and Medical Oncology, Department of Internal Medicine, Dr Sardjito General Hospital, Yogyakarta, Indonesia, 4 Department of Anatomical Pathology, Faculty of Medicine, Public Health and Nursing, Universitas Gadjah Mada/ Dr Sardjito General Hospital, Yogyakarta, Indonesia, 5 Division of Hematology and Medical Oncology, Department of Internal Medicine, Faculty of Medicine, Public Health and Nursing, Universitas Gadjah Mada/ Dr Sardjito General Hospital, Yogyakarta, Indonesia

* susanna.hutajulu@ugm.ac.id

## Abstract

### Objectives

To observe pre- and post-treatment vitamin D level and its association with treatment and concomitant factors in breast cancer patients treated with chemotherapy.

### Methods

We performed a pre-post observational analysis that nested in an ongoing prospective cohort study of breast cancer patients at Dr. Sardjito General Hospital, Yogyakarta, Indonesia. 136 subjects were recruited from the main study. Information on subjects' socio-demographic characteristics clinical status, and tumour profile was assessed at baseline. Number of chemotherapy cycles and chemotherapy-induced nausea vomiting (CINV) were also recorded. Vitamin D concentration was measured using ELISA methods at baseline and post-treatment. Vitamin D level of <20 ng/ml and <12 ng/ml were defined as deficiency and severe deficiency. Correlation between socio-demographic and clinical profile with baseline vitamin D was tested using Spearman correlation. Paired t-test was used to evaluate changes in post-treatment vitamin D concentration. The odds ratio for a subject to experience post-treatment vitamin D decrease was assessed based on number of chemotherapy cycles and CINV severity.

### Results

The mean vitamin D level before chemotherapy was very low (8.80±3.64 ng/ml) in the whole panel. Higher AST level were associated with lower vitamin D level at baseline (r = -0.188, p

the ethics committee as most of these contain patient data, albeit de-identified, and it may be possible to determine the identity of participants given the extent of sociodemographic and clinical data available for each participant. We provided a minimal data set as Supporting Information that included deidentified data used to generate all tables. Should there be a request for data, this can be sent to the corresponding author (email: susanna.hutajulu@ugm.ac.id). Future researchers can contact the institutional ethics committee (email: mhrec_fmugm@ugm.ac.id) at Universitas Gadjah Mada, Indonesia, with data access queries as well.

**Funding:** SHH received funding from Kementrian Riset, Teknologi dan Pendidikan Tinggi, Republik Indonesia (ID) (2018) and Universitas Gadjah Mada (2020). We confirm that our study only received funding from the stated institutions and the funders had no role in the study design, data collection and analysis, decision to publish, or preparation of the manuscript.

**Competing interests:** The authors have declared that no competing interests exist.

= 0.028). Severe deficiency was found in 82.4% subjects at baseline and the rate increased to 89.0% after chemotherapy. Eighty-five cases showed a decrease level whereas 51 showed a slight improvement. Overall, a significant decrease of the vitamin D level was observed after chemotherapy (median change 3.13±4.03 ng/ml, p <0.001). Subjects who received >6 cycles of chemotherapy were less likely to experience a decreased level of post-treatment vitamin D (OR = 0.436, 95% CI = 0.196–0.968, p = 0.039).

## Conclusions

Indonesian breast cancer patients showed pre-existing severe vitamin D deficiency and deterioration of vitamin D after chemotherapy. Future research is needed to explore its implication towards patients' survival in the local setting. Evidence-based approach also needs to be taken to address this modifiable condition, including increasing awareness of the importance of maintaining vitamin D sufficiency both in patients and the general population.

## Introduction

Breast cancer remains the most frequent cancer in women with 2.1 million new cases recorded in 2018 [1]. It is also the most frequent cancer and the second-highest cause of cancer-related mortality in Indonesia with 44.0 incidences and 15.3 death per 100.000 population [2]. Cases in Indonesia were more often present in younger patients and diagnosed at an advanced stage (IIIA-C and IV), contributing to the low 5-year survival rate (51.1%) [3–5].

New evidence has proven to be associated with breast cancer survival, including vitamin D. Poor vitamin D concentration was associated with poor breast cancer prognostic characteristics as lower vitamin D concentration was found in patients with poor differentiation, stage 4, and negative ER expression [6]. On the contrary, maintained vitamin D concentration was associated with a better clinical outcome due to its ability to inhibit abnormal cell growth and differentiation and to regulate local vitamin D synthesis in breast tissue [7–9]. Supporting evidence also observed that ≥23.6 ng/ml pre-diagnostic vitamin D concentration on newly diagnosed patients has 59% lower risk of breast cancer-related mortality Moreover, improvement of vitamin D was also correlated with lower risk of fatality (HR = 0,57, 95% CI = 0,43–0,75) [10].

Studies have observed that vitamin D levels might deteriorate due to chemotherapy, the treatment modality widely used in Indonesia [11, 12]. Chemotherapy regimens such as anthracycline, cyclophosphamide, and taxane are known to cause gonad toxicity and resting oocytes destruction, reducing estrogen expression and increasing vitamin D catabolism [13, 14]. Additionally, concomitant effects of chemotherapy were associated with reduced sun exposure, physical activity, and diet limitation, further adding to the low vitamin D level [15, 16].

To date, observation of vitamin D concentration in the Indonesian population was focusing on the population at risks such as children, adolescents, and pregnant women [17–19]. Although observation of vitamin D concentration in postmenopausal breast cancer patients has been done previously in Surabaya, Indonesia [20], the association between vitamin D level and breast cancer characteristics remains unexplored. No study evaluating post-treatment vitamin D changes has ever been done. Herein, we are aiming to observe vitamin D changes in Indonesian breast cancer patients treated with chemotherapy and its association with chemotherapy and concomitant factor.

## Methods

### Study design and participants

We performed a nested pre-post observational study assessing 25(OH)D changes on primary breast cancer patients registered in a prospective ongoing cohort. The main study analysed the risk of chemotherapy side effects and its effect on survival and quality of life in breast cancer patients. Subjects of the main study were histologically confirmed female breast cancer patients aged ≥18 years who were chemotherapy naïve and receiving their first chemotherapy in the Haematology and Medical Oncology Division, "Tulip"/Integrated Cancer Clinic, Dr. Sardjito General Hospital, Yogyakarta, Indonesia, from 2018–2022. No subjects in the cohort were in a terminal condition or with severe congestive heart failure. Among subjects enrolled in the main cohort, we only included those who have received their last chemotherapy cycle and completed post-treatment follow-up into the present study. Subjects who have incomplete blood sample were excluded. The main study was approved by the Medical and Health Research Ethics Committee of the Faculty of Medicine, Public Health and Nursing, Universitas Gadjah Mada/Dr. Sardjito Hospital (reference number: KE/FK/0417/EC/2018) and was amended for the study reported in this manuscript (amendment approval with reference number: KE/FK/0432/EC/2020). Written informed consent was obtained before subjects' enrolment.

### Clinical data, breast cancer characteristics, and follow-up

Demographic and clinical data, including age, menarche, menopausal status, parity, education, occupation, insurance, BMI, and upper arm circumference was collected from the main study database. Baseline haemoglobin (Hb), albumin, leucocyte, aspartate transaminase (AST), and alanine transaminase (ALT) concentration measurements were performed within one week prior to subjects' first chemotherapy cycle. Clinical staging was determined using the 7th edition of the American Joint Committee on Cancer (AJCC). Details on tumour profile and biomarkers expression were obtained through the patient's histopathological examination result. Estrogen receptor (ER), progesterone receptors (PR), and human epidermal growth factor 2 (HER2) was defined according to the American Society of Clinical Oncology/College of American Pathologist (ASCO/CAP) guidelines [21, 22].

Subjects received chemotherapy as neoadjuvant treatment (before surgery), adjuvant treatment (after surgery), or palliative treatment with or without surgery, as planned by their respective oncologist. Most subjects in this study received AC-T (Doxorubicin-Cyclophosphamide-Taxane, 47.8%), EC-T (Epirubicin-Cyclophosphamide-Taxane, 11.8%), or FEC-T (5-Fluorouracyl-Epirubicin-Cyclophosphamide+Taxane, 10.3%) chemotherapy regimen. Ten subjects received TCb (Taxane-Carboplatin, 7,4%) chemotherapy regimen. Capecitabine was given for 10 subjects in total (7.4%), seven (5.2%) as single-agent therapy, two (1.5%) following Taxane, while one (0.8%) was given following Taxane and in combination with cyclophosphamide. Herceptin was used in the treatment of three subjects in total (2.2%), each following AC-T, EC-T, and FEC-T.

Follow-up was conducted within 2 weeks after the subjects' last chemotherapy cycle as part of observation in the main study to evaluate the total chemotherapy cycles received. Chemotherapy-induced nausea and vomiting (CINV) episodes were recorded on each chemotherapy cycle evaluation. Based on the median of CINV episodes experienced by subjects, CINV was categorized as mild if one experienced it in no more than 4 treatment cycles and severe otherwise.

## Vitamin D measurements

Vitamin D concentrations were measured from blood samples taken at baseline and post-chemotherapy follow-up using standard phlebotomy procedures. Blood samples were collected within one week prior to the date of first chemotherapy administration (baseline) and after the last chemotherapy dose (post-treatment), where possible. Covid 19 pandemic affected the timeliness of the biological sample collections, causing altered schedule in the blood sample taking, both at baseline and post-treatment observation points (ranged 0–147 days and 3–125 days, respectively). After being centrifugated, plasma samples were aliquoted and stored at -80˚C until analysis. Prior to vitamin D measurements, no sample has undergone more than 2 freeze-thawing cycles. Vitamin D concentration was measured as 25(OH)D with ELISA method using DRG 25(OH)-Vitamin D kit (DRG, Marburg, Germany; Cat no. EIA-5396. The intra- and inter-assay coefficient of variability (%CV) was 4.7% and 10.2%, respectively. Sensitivity of the assay was 74.7% for 25-OH-Vitamin D2 and 100% for 25-OH-Vitamin D3. Sample storage and analysis were done at the Biobank and the Integrated Research Laboratory of the Faculty of Medicine, Public Health, and Nursing, Universitas Gadjah Mada, performed by research team member blinded to subjects' personal and clinical characteristics. Vitamin D concentration was defined as sufficient ($\geq$30 ng/ml), insufficient (20.0–29.9 ng/ml), deficient (12.0–19.9 ng/ml), and severely deficient (<12.0 ng/ml) [23–25].

## Statistical analysis

It is estimated that at least 119 samples are needed to achieve 90% power and 5% two-sided level of significance to observe 2ng/mL mean difference in vitamin D concentration, assuming the standard deviation of the difference to be 6.67 ng/ml [26]. Kolmogorov-Smirnov test was performed to observe the distribution of all numeric variables. Demographic and clinical characteristics at baseline were explored using descriptive analyses. Independent t-test and ANOVA were used to observe differences of mean vitamin D among subjects with different characteristics. Independent t-test was also performed to compare mean vitamin D based on the duration of sample collection and the season when the sample was collected. The season was determined based on the date of sample collection and grouped into dry and rainy seasons according to the report of the national Meteorological, Climatological, and Geophysical Agency. The Spearman correlation analysis was employed to observe the association between baseline vitamin D concentration with the demographic and clinical profile. Missing data were excluded from bivariate analysis for the corresponding factor. Paired t-test was then performed to evaluate the changes in vitamin D concentration between baseline and post-chemotherapy. The odds ratio for a subject to experience a decrease in their post-chemotherapy vitamin D concentration was analysed based on the variance in the number of chemotherapies cycles and concomitant effect during treatment, namely CINV. P-value of the odds for chemotherapy duration and CINV severity was reported based on Pearson's chi-square test. Shorter chemotherapy duration and less-severe CINV were used as the reference group. All analyses were performed with IBM SPSS Statistics 17.0 (IBM Inc., Armonk, USA) and a two-sided p-value of $\leq$0.05 was considered statistically significant.

## Results

### Characteristics of the study population

A total of 212 women who met the inclusion criteria have been registered in the main study by June 2021. Among them, 44 were still undergoing chemotherapy programs and thus were not included in the present study. A further 32 subjects were excluded because none of their post-

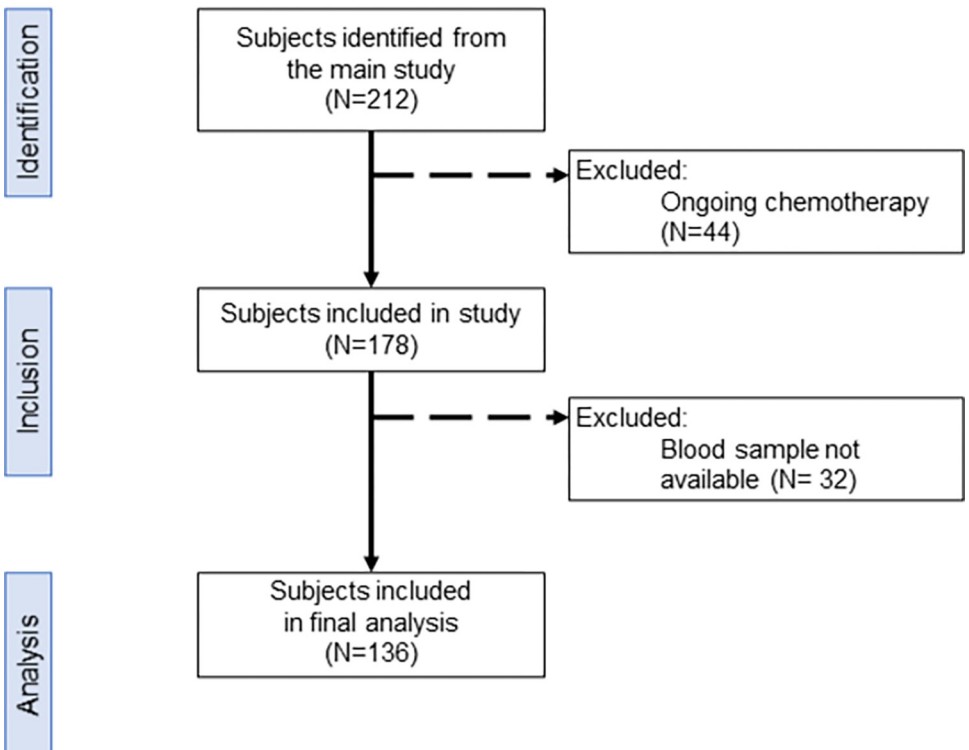

**Fig 1. Flowchart of subjects' inclusion process.** Flowchart showed subjects' identification from the parenting study. Of the 212 participants already recruited in the main study, 44 were not included because they had not yet completed all planned chemotherapy cycles. Thirty-two participants were excluded because their blood samples were not available.

chemotherapy blood samples were available, making a total number of 136 patients included in the nested study (Fig 1).

The median age of the subjects was 51 years old (Table 1). The median age of subjects' menarche was 14 years old and more than half were already menopause by the time of the study (72, 52.9%). Among 67 (93.1%) subjects who provided information of their menopause, the median age of menopause is 50±6 years. The median duration of menopause is 7.91±10.5 years at enrolment. Aside from the seventeen subjects who had never given birth (17, 12.5%), most were multiparous (104, 76.5%). Most subjects were well educated (74, 54.4%), are working (74, 54.4%), and owned paid or private insurance (111, 81.6%). Subjects have a good clinical profile at enrolment, with 96.3% presented with ECOG 1 and the median value of all clinical parameters was observed within the normal range at baseline. Although underweight subjects dominated the study (51, 37.5%), the median BMI was within the normal category (20.91 kg/m2). Most subjects were diagnosed with ductal infiltrative breast cancer (115, 84.6%) and have poor histological differentiation (82, 60.3%). Most subjects were presented with locally advanced breast cancer (60, 44.1%). Subjects' immunohistochemistry profiles showed that more subjects have positive ER and PR expression (81, 59.6% and 69, 50.7%, respectively), and negative HER2 expression (90, 66.2%).

One-hundred and twenty-two subjects (89.0%) have undergone surgery prior to the first cycle of chemotherapy. The median duration of chemotherapy was 5 months with 12 (8.8%) subjects receiving treatment for longer than 6 months. Of all subjects, 101 (74.3%) received a combination of anthracycline, cyclophosphamide, and taxane during treatment.

**Table 1. Baseline characteristics of study subjects (n = 136).**

| Baseline characteristics | Frequency N (%) |
|---|---|
| Age (years; median±IR) | 51.00±13.00 |
| Age of menarche (years; median±IR) | 14.00±3.00 |
| Menopause | |
| Menopause | 72 (52.9) |
| Pre-menopause | 64 (47.1) |
| Age of first pregnancy (years; median±IR) | 23.50±8.00 |
| Parity | |
| Nullipara | 17 (12.5) |
| Primipara | 15 (11) |
| Multipara | 104 (76.5) |
| Education | |
| Undereducated | 62 (45.6) |
| Well educated | 74 (54.4) |
| Occupation | |
| Housewives | 62 (45.6) |
| Workers | 74 (54.4) |
| Insurance | |
| Underprivileged insurance | 25 (18.4) |
| Paid and private insurance | 111 (81.6) |
| BMI (kg/m$^2$; median±IR) | 20.91±10.63 |
| Underweight-normal (<25.00 kg/m2) | 99 (72.8) |
| Overweight-obese (≥25.00 kg/m$^2$) | 37 (27.2) |
| Upper arm circumference (cm; median±IR) | 27.00±6.00 |
| Hb (mg/dL; mean±SD) | 12.65±1.85 |
| Albumin (g/dL; median±IR) | 4.51±0.74 |
| Leucocyte (x10$^6$/ul; median±IR) | 7.09±2.23 |
| AST (IU/l; median±IR) | 20.00±11.00 |
| ALT (IU/l; median±IR) | 23.00±14.00 |
| Histological type | |
| Ductal infiltrative | 115 (84.6) |
| Lobular infiltrative | 9 (6.6) |
| Others | 12 (8.8) |
| Grade | |
| Well-moderately differentiated | 34 (25) |
| Poorly differentiated | 82 (60.3) |
| Missing | 20 (14.7) |
| Tumor size | |
| ≤T1 | 13 (9.6) |
| T2 | 39 (28.7) |
| T3 | 42 (30.9) |
| T4 | 42 (30.9) |
| Stage | |
| Stage 1–2 | 43 (31.6) |
| Stage 3 | 60 (44.1) |
| Stage 4 | 33 (24.3) |
| ER | |
| Positive | 81 (59.6) |

(*Continued*)

**Table 1.** (Continued)

| Baseline characteristics | Frequency N (%) |
|---|---|
| Negative | 53 (39) |
| Missing | 2 (1.5) |
| PR | |
| Positive | 69 (50.7) |
| Negative | 65 (47.8) |
| Missing | 2 (1.5) |
| HER2 | |
| Positive | 41 (30.1) |
| Negative | 90 (66.2) |
| Missing | 5 (3.7) |
| Number of chemotherapy cycles | |
| ≤6 cycles | 43 (31.6) |
| >6 cycles | 93 (68.4) |
| CINV | |
| None-mild | 76 (55.9) |
| Severe | 60 (44.1) |
| Baseline vitamin D concentration (ng/ml; median±IR) | 8.44±4.59 |
| Sufficient | 0 (0) |
| Insufficient | 1 (0.7) |
| Deficient | 23 (16.9) |
| Severely deficient | 112 (82.4) |

Abbreviation: IR: interquartile range; BMI: body mass index; ALT: alanine transaminase; AST: aspartate transaminase; ER: estrogen receptor; PR: progesterone receptor; HER2: human epidermal growth factor receptor 2; ACT: anthracycline-cyclophosphamide-taxane.

Chemotherapy regimens were given in more than 6 cycles in 93 subjects (68.4%). After evaluating the concomitant effect experienced by subjects during treatment, eight of them never experience any nausea or vomiting while mild and severe CINV was more common (68, 50% and 60, 44.1%, respectively).

## Vitamin D profile at baseline

In this study, none of the subjects has normal vitamin D status. Median vitamin D at baseline was 8.44±4.59 ng/ml and was below the cut-off of severe deficiency on different patients' and tumours' characteristics (Tables 1 and 2). We found no significant difference in mean vitamin D levels among subjects with different menopausal status, parity, education, occupation, insurance, BMI, and tumour characteristics (S1 Table). Based on the median of age (51 years old), median baseline vitamin D is similar among younger and older subjects (p = 0.428). Median vitamin D also did not differ (p = 0.819) among subjects who were underweight (median±IR = 8.06±5.44 ng/ml), normal (median±IR = 9.10±5.27 ng/ml), overweight (median±IR = 9.68±3.16 ng/ml), and obese (median±IR = 8.23±2.82 ng/ml). A significant correlation was observed between baseline vitamin D concentration with AST level (r = -0.188, p = 0.028) (Table 2). Vitamin D concentration was not correlated with socio-demographic and other clinical profiles.

## Post-chemotherapy vitamin D changes

A lower median vitamin D concentration was observed post-treatment (median±IR = 6.89 ±4.44 ng/ml) (Table 3). The paired t-test analysis found a significant change in the vitamin D

**Table 2. Correlation between baseline vitamin D level with socio-demographic factors and clinicopathology characteristics (n = 136).**

| Predictors | Baseline vitamin D[a] (ng/ml; median±IR) | Spearman Correlation | |
|---|---|---|---|
| | | R | p-value |
| Age (years; median±IR) | | 0.096 | 0.266 |
| Age of menarche (years; median±IR) | | 0.005 | 0.955 |
| Menopause | | 0.086 | 0.317 |
| Menopause | 8.62±5.33 | | |
| Pre-menopause | 8.18±4.46 | | |
| Age of first pregnancy (years; median±IR) | | -0.012 | 0.895 |
| Parity | | -0.015 | 0.866 |
| Nullipara | 9.78±6.02 | | |
| Primipara | 5.69±6.11 | | |
| Multipara | 8.43±3.85 | | |
| Education | | 0.059 | 0.496 |
| Undereducated | 8.21±3.50 | | |
| Well educated | 8.67±5.45 | | |
| Occupation | | 0.003 | 0.974 |
| Housewives | 8.45±4.23 | | |
| Workers | 8.49±4.89 | | |
| Insurance | | 0.030 | 0.729 |
| Underprivileged insurance | 8.06±4.73 | | |
| Private insurance | 8.50±4.51 | | |
| BMI (kg/m$^2$; median±IR) | | 0.083 | 0.336 |
| Underweight-normal (<25.00 kg/m2) | 8.45±5.31 | | |
| Overweight-obese (≥25.00 kg/m2) | 8.43±2.84 | | |
| Upper arm circumference (cm; median±IR) | | 0.034 | 0.699 |
| Hb (mg/dL; mean±SD) | | -0.062 | 0.476 |
| Albumin (g/dL; median±IR) | | -0.095 | 0.308 |
| Leucocyte (x10$^6$/ul; median±IR) | | -0.005 | 0.959 |
| AST (IU/l; median±IR) | | -0.188 | 0.028[b] |
| ALT (IU/l; median±IR) | | -0.144 | 0.096 |
| Histological type | | -0.040 | 0.644 |
| Ductal infiltrative | 8.45±4.91 | | |
| Lobular infiltrative | 3.74±5.73 | | |
| Others | 7.87±5.35 | | |
| Grade | | -0.051 | 0.585 |
| Well-moderately differentiated | 8.79±5.56 | | |
| Poorly differentiated | 8.26±4.72 | | |
| Tumor size | | 0.124 | 0.149 |
| ≤T1 | 8.04±5.68 | | |
| T2 | 7.94±4.53 | | |
| T3 | 8.12±4.82 | | |
| T4 | 9.21±4.31 | | |
| Stage | | 0.025 | 0.776 |
| Stage 1–2 | 8.43±3.43 | | |
| Stage 3 | 8.31±4.93 | | |
| Stage 4 | 9.68±6.41 | | |
| ER | | -0.110 | 0.205 |
| Positive | 8.42±4.77 | | |
| Negative | 8.51±5.16 | | |
| PR | | -0.063 | 0.473 |

*(Continued)*

**Table 2.** (Continued)

| Predictors | Baseline vitamin D$^a$ (ng/ml; median±IR) | Spearman Correlation | |
|---|---|---|---|
| | | R | p-value |
| Positive | 8.43±4.64 | | |
| Negative | 8.51±4.88 | | |
| HER2 | | 0.106 | 0.227 |
| Positive | 8.59±4.11 | | |
| Negative | 8.43±4.88 | | |

Abbreviation: IR: interquartile range; 95% CI: 95% confidence interval; BMI: body mass index; ALT: alanine transaminase; AST: aspartate transaminase; ER: estrogen receptor; PR: progesterone receptor; HER2: human epidermal growth factor receptor 2.

Due to missing values, the counts of some variables did not add up to the total.

$^a$Independent t-test and ANOVA test were done to compare median vitamin D levels and no significant difference was found among groups. Results are presented in S1 Table.

$^b$Statistically significant.

concentration, as compared to the level measured at baseline (p <0.001). The median and interquartile value obtained from computing the absolute changes of vitamin D between the two observation points was 3.13±4.03 ng/ml. Even though the number of severely deficient subjects was already high at baseline (82.4%), the proportion rose to 89.0% after chemotherapy. Sixteen subjects (69.6%) who were deficient and the only subject who was insufficient at baseline become severely deficient by the end of observation. Although an increase in vitamin D concentration was observed in 51 subjects (37,5%), only 7 (6.25%) showed status improvement from severely deficient to deficient.

The median duration between sample collection with first chemotherapy was 8±7 days and 13±22 days with the last chemotherapy dose administration (S2 Table). Distance of blood sample collection was more than 14 days in 26 subjects (19.18%) at baseline and 62 subjects (45.60%) at post-treatment observation. On both observation points, we found no significant difference in vitamin D level between samples taken within 14 days and more than 14 days from the closest chemotherapy administration (S3 Table).

Seventy-one baseline samples (52.21%) and 78 post-treatment samples (57.35%) were taken in the rainy season. Median vitamin D level was similar between samples taken in the dry and rainy season, both in baseline (median±IR = 8.02±5.36 vs. 9.09±3.50 ng/ml, p = 0.078) and post-treatment observation (median±IR = 7.46±5.01 vs. 6.45±4.54 ng/ml, p = 0.344) (S4 Table).

## Association between post-chemotherapy vitamin D level and chemotherapy factors

After chemotherapy, vitamin D concentration was decreased on 84 subjects (61.8%) (Table 4). Among them, most received treatment for less than 6 months (75, 89.3%) and received more

**Table 3.** Vitamin D concentration changes between baseline and post-chemotherapy in breast cancer patients (n = 136).

| Time point | Vitamin D concentration | Paired T-test p-value |
|---|---|---|
| | (ng/ml; median±IR) | |
| Baseline | 8.44±4.59 | p <0.001 |
| Post-treatment | 6.89±4.44 | |

Abbreviation: IR: interquartile ratio; COV: coefficient of variability.

**Table 4. Association of chemotherapy cycles and nausea and vomiting with the decrease of post-treatment vitamin D concentration (n = 136).**

| Variables | Post-treatment vitamin D[a] | | | | Pearson's Chi Square | |
|---|---|---|---|---|---|---|
| | Decrease | | Increase/persist | | OR (95% CI) | p-value |
| | n | % | n | % | | |
| **Chemotherapy cycles** | | | | | | |
| ≤6 cycle | 32 | 38.1 | 11 | 21.2 | 1.000 | 0.039[b] |
| >6 cycle | 52 | 61.9 | 41 | 78.8 | 0.436 | |
| | | | | | (0.196–0.968) | |
| **CINV** | | | | | | |
| None-mild | 51 | 60.7 | 25 | 48.1 | 1.000 | 0.149 |
| Severe | 33 | 39.3 | 27 | 51.9 | 0.599 | |
| | | | | | (0.298–1.204) | |

Abbreviation: OR: Odd ratio; 95% CI: 95% confidence interval; CINV: chemotherapy-induced nausea and vomiting.

[a]Independent t-test and ANOVA test were done to compare median vitamin D levels and no significant difference was found among subjects with different chemotherapy cycles and CINV severity groups. Results are presented in S5 Table.

[b]Statistically significant.

than six chemotherapy cycles (n = 52, 61.9%). Vitamin D decrease was observed in 33 subjects who experienced severe CINV (39.3%) and 51 who experienced mild or did not experience CINV (60.7%). No significant difference was observed in the vitamin D level between different chemotherapy cycles and CINV severity (p = 0.381 and 0.338 respectively) (S5 Table).

We observed that subjects who received more than 6 cycles of chemotherapy were less likely to have worse post-treatment vitamin D compared to those who received 6 cycles or less (OR = 0.436, 95% CI = 0.196–0.968, p = 0.039). Vitamin D level was less likely to decrease in subjects who experience severe CINV (OR = 0.599, 95% CI = 0.298–1.204, p = 0.149).

## Discussion

### Summary of key findings

This is the first study evaluating pre- and post-treatment vitamin D changes in Indonesian breast cancer patients and exploring factors associated with it. We found that Indonesian breast cancer patients have very low vitamin D concentration and experience a significant deterioration after chemotherapy. Among socio-demographic and clinical factors observed in our study, AST was the only factor associated with vitamin D concentration. Moreover, we discovered that patients who had more chemotherapy cycles were less likely to experience a decrease in their post-treatment vitamin D, emphasising the significance of chemotherapy effects towards vitamin D concentration in already deficient subjects.

### Comparison of vitamin D profile with previous studies

The high prevalence of vitamin D deficiency (VDD) and low vitamin D concentration in breast cancer patients have been observed, both in European, Australian and Asian populations (mean ranging from 12.4 to 22.8 ng/ml) [6, 12, 27–29]. Post-treatment vitamin D deterioration has also been observed in previous studies in other populations [11, 12]. However, the median vitamin D concentration observed in this study was lower than previously reported.

Information on vitamin D status in the Indonesian general population are very limited and have only been assessed for the certain population at risk, such as pregnant women, newborn, children, and, as recently observed, among adult Covid-19 patients [17–19, 30, 31]. Although

studies of vitamin D profile in Indonesian cancer patients remain limited, previous studies have highlighted the low vitamin D concentration (mean = 21.2 and 15.7 ng/ml) and high prevalence of VDD (60%) in healthy Indonesian women [32–34]. In accordance, our study shows very low vitamin D concentration at baseline with VDD observed in almost all subjects (99.4%), despite having good clinical profiles.

In addition, low vitamin D concentration was observed regardless of menopausal status, parity, education level, occupation, and insurance in this study. This result suggests the pre-existing poor vitamin D concentration in the general population. Moreover, this study also does not find the existence of seasonal influence toward subjects' vitamin D status. Although Indonesia's geographical location ensures year-round sun exposure, the climate, and high sun exposure encourage sun-protective behaviour, limiting UV B exposure and vitamin D synthesis [12, 34, 35]. Moreover, the Indonesian population has a tendency to dress modestly, covering most of the skin. Along with cultural belief to avoid sun exposure and skin tanning, this may lead to a lower vitamin D concentration in Indonesian women [35, 36].

Despite Indonesia and most of the South East Asian countries being thought to have a low risk of VDD due to its location near the equator, recent studies show otherwise [17, 35–37]. The lower vitamin D concentration was not only observed in the population living in South East Asia but also of its descent who live in Europe (mean = 22.1 ng/ml and 22.8 ng/ml, respectively). This suggests an influence of genetic variance on vitamin D metabolism that warrants further investigation [38, 39]. Aside from the difference in skin tone, the higher visceral adipose tissue (VAT) in South Asian compared to European descent is associated with higher sequestration and therefore lower vitamin D concentration [40]. The higher proportion of VAT was found in Asian population despite of lower BMI [41–43]. The stronger negative correlation between VAT with vitamin D level instead of the overall body fat might explain the low vitamin D across different BMI categories as observed in this study [40].

Vitamin D concentration observed in our study was lower than previously observed in locally advanced breast cancer women in Surabaya, Indonesia (median = 17.41 ng/ml) [20]. The previous study was done in smaller sample size and only include post-menopausal locally advanced breast cancer patients receiving CAF neoadjuvant treatment. Although the different patients' characteristics presented in our study might provide vitamin D profile of Indonesian breast cancer patients in a wider scope, we were also unable to dismiss the possibility of different diets and sun behaviour between the two study samples also influencing the lower vitamin D status presented in this study.

The vitamin D measurement assay selected for this study have high sensitivity and specificity, and good comparability to similar vitamin D assays [44, 45]. All samples were kept in -80˚C prior to analysis with no history of freeze-thaw. All samples, QC low, QC high, and standards from the kits were treated uniformly and concurrently, minimising any possible treatment biases of the assay. The vitamin D concentration of the samples falls within the range of the standard curve and low measurement results were not influenced by the type of assay used. Considering the similarly low vitamin D concentration observed in Korean breast cancer patients and the high prevalence of deficiency in various studies on different at-risk population in Indonesia, the very low vitamin D concentration observed in our study suggest the severity of vitamin D deficiency in Indonesian women [17, 18, 29, 30].

The only variable significantly correlate with vitamin D level at baseline was AST serum level. All subjects enrolled in the study had a good performance and clinical status, and none had any form of liver disorder at the beginning of the study. Despite median AST concentration being within the normal range, we found that higher AST was associated with lower vitamin D concentration. Although the association between vitamin D and AST has not been widely investigated in breast cancer patients, several studies have explored its correlation with

liver disease. In line with our discovery, higher vitamin D concentration was associated with lower odds of higher AST in previous study (OR = 0.97, 95% CI = 0.93–1.00, p <0.05) [46]. While the exact mechanism needs to be further explored, increased AST concentration was associated with a disturbance of liver function, causing decreased expression of D binding protein and ineffective gastrointestinal vitamin D absorption, and in turn, decreased vitamin D concentration [8, 47–49].

Our study observed that subjects who received more than 6 chemotherapy cycles had lower odds to experience vitamin D decrease compared to patients who were only prescribed 6 cycles or less. Although more chemotherapy cycle does not necessarily mean a longer treatment duration, a study performed in Australian breast cancer women also find that patients' mean vitamin D level was lower at 6th week observation compared to 12th week [12]. Chemotherapy, especially anthracycline, cyclophosphamide, and taxane which were mainly used in the treatment of subjects in this study, are known to cause gonad toxicity and resting oocytes destruction. This will in turn cause reduced estrogen expression and increased vitamin D catabolism [13, 50]. Vitamin D decreases have been observed in breast cancer patients receiving ACT, AC, and FEC-T [11, 14]. The chemotherapy regimen in this study is quite homogenous as treatments were given in accordance with the standard therapy regimen based on tumour characteristics. The high-frequency anthracycline and taxane used might cause the contradicting results observed.

Although not statistically significant, our study suggests that more severe CINV was associated with higher post-chemotherapy vitamin D concentration. CINV in cancer patients was associated with poor intake and worse nutritional status [51]. However, the correlation between CINV severity and vitamin D concentration as observed in this study could be explained by diet modification, as commonly encouraged in the local practice. A similar result was also observed by others [12]. Consumption of fortified food and supplement are often encouraged to help patients fulfil their nutritional needs despite intake limitation caused by CINV. Moreover, a previous preclinical study [52] shows higher intestinal absorption and hydroxylation in vitamin D deficient, suggesting a compensating mechanism might occur in low vitamin D concentration.

## Clinical and population health applicability

Maintained vitamin D concentration was associated with a better clinical outcome in breast cancer patients. A previous study discovered that improvement of vitamin D was correlated with a lower risk of fatality (HR = 0.57, 95%CI = 0.43–0.75) [10]. This is due to its ability to inhibit abnormal cell growth and differentiation and to regulate local vitamin D synthesis on breast tissue [7–9]. To perform this function, vitamin D is needed at its normal concentration at 30 ng/ml [7].

One way to maintain and improve vitamin D sufficiency was through supplementation [53]. Nevertheless, it is essential to provide vitamin D supplementation in adequate dosage to avoid ineffective treatment due to unmet therapeutic dose [54] or toxicity due to excess dosage [15]. The low mean vitamin D concentration observed in our study suggests that a higher dose of vitamin D supplementation might be necessary. A higher dose of vitamin D supplementation has shown superior results in normalizing vitamin D concentration (30% vs 12.6% improvement, p = 0.003) [55]. Although the maximum dose of 10,000 IU/day has been found to be the no-observed-adverse-effect level (NOAEL), caution must be taken concerning the J-shaped relationship between vitamin D concentration and breast cancer prognosis as observed on vitamin D concentration above 44 ng/ml (HR = 1.63, 95% CI = 1.21–2.19) [7, 54]. Further study is warranted to determine the most suitable dose, duration, and preparation for Indonesian breast cancer patients.

## Study strength and limitation

The strength of the study was the fact that it was conducted in a tertiary hospital in Yogyakarta, the region with the highest cancer prevalence in the country [56] where breast cancer was the most common [57, 58]. Thus, it was expected to be able to represent the regional breast cancer population. However, some limitations should be considered in the interpretation of our results. Sun behaviour, clothing coverage, skin tone, body fat distribution, and daily vitamin D intake were not observed and thus warrant further exploration. In addition, the chemotherapy used by subjects of our study were commonly prescribed in local practice. Further study with matching control and controlled chemotherapy variant should be done to elucidate the effect of different regimens on vitamin D changes.

## Conclusion

Baseline vitamin D is very low in Indonesian breast cancer patient and decrease significantly afterward. Vitamin D concentration in breast cancer patients is associated with AST level while post-treatment vitamin D changes is associated with chemotherapy cycle. Although further study is needed to explore the dynamic of vitamin D changes during chemotherapy, treatment outcome, and the optimal vitamin D supplementation needed in the local population, our study provides important data to improve care for breast cancer patients, especially those receiving chemotherapy.

## Supporting information

**S1 Table. Comparison of baseline vitamin D level among different socio-demographic factors and clinicopathology characteristics (n = 136).** Abbreviation: IR: interquartile range; BMI: body mass index; ER: estrogen receptor; PR: progesterone receptor; HER2: human epidermal growth factor receptor 2.
(PDF)

**S2 Table. Duration between sample collection and nearest chemotherapy administration (n = 136).** Abbreviation: IR: interquartile range.
(PDF)

**S3 Table. Comparison of vitamin D level among different distances of sample collection from nearest chemotherapy administration (n = 136).** Abbreviation: IR: interquartile range.
(PDF)

**S4 Table. Comparison of vitamin D level among different seasons at point of sample collection (n = 136).** Abbreviation: IR: interquartile range.
(PDF)

**S5 Table. Comparison of post-treatment vitamin D level among different chemotherapy factors (n = 136).** Abbreviation: IR: interquartile range; CINV: chemotherapy-induced nausea vomiting.
(PDF)

**S1 File. Minimal data set.**
(XLSX)

## Acknowledgments

The authors thank Irfan Haris, Yufi Kartika Astari, Norma Dewi Suryani, Betrix Rifana Kusumaning Indah, and Sumartiningsih for technical assistance and coordination.

## Author Contributions

**Conceptualization:** Herindita Puspitaningtyas, Dian Caturini Sulistyoningrum, Susanna Hilda Hutajulu.

**Data curation:** Herindita Puspitaningtyas, Susanna Hilda Hutajulu.

**Formal analysis:** Herindita Puspitaningtyas, Dian Caturini Sulistyoningrum, Susanna Hilda Hutajulu.

**Funding acquisition:** Herindita Puspitaningtyas, Dian Caturini Sulistyoningrum, Susanna Hilda Hutajulu.

**Investigation:** Herindita Puspitaningtyas, Dian Caturini Sulistyoningrum, Riani Witaningrum, Susanna Hilda Hutajulu.

**Methodology:** Herindita Puspitaningtyas, Dian Caturini Sulistyoningrum, Irianiwati Widodo, Mardiah Suci Hardianti, Susanna Hilda Hutajulu.

**Project administration:** Herindita Puspitaningtyas, Riani Witaningrum.

**Resources:** Herindita Puspitaningtyas, Dian Caturini Sulistyoningrum, Riani Witaningrum, Mardiah Suci Hardianti, Kartika Widayati Taroeno-Hariadi, Johan Kurnianda, Ibnu Purwanto, Susanna Hilda Hutajulu.

**Supervision:** Dian Caturini Sulistyoningrum, Irianiwati Widodo, Mardiah Suci Hardianti, Kartika Widayati Taroeno-Hariadi, Johan Kurnianda, Ibnu Purwanto, Susanna Hilda Hutajulu.

**Validation:** Herindita Puspitaningtyas, Dian Caturini Sulistyoningrum, Riani Witaningrum, Susanna Hilda Hutajulu.

**Visualization:** Herindita Puspitaningtyas, Dian Caturini Sulistyoningrum, Susanna Hilda Hutajulu.

**Writing – original draft:** Herindita Puspitaningtyas, Dian Caturini Sulistyoningrum, Susanna Hilda Hutajulu.

**Writing – review & editing:** Herindita Puspitaningtyas, Dian Caturini Sulistyoningrum, Riani Witaningrum, Irianiwati Widodo, Mardiah Suci Hardianti, Kartika Widayati Taroeno-Hariadi, Johan Kurnianda, Ibnu Purwanto, Susanna Hilda Hutajulu.

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
