## [Editor Report · Decision Letter 0]

10 Feb 2022

PONE-D-21-36153Vitamin D status in breast cancer patients following chemotherapy: a pre and post observational study in a tertiary hospital in Yogyakarta, IndonesiaPLOS ONE

Dear Dr. Hutajulu,

Thank you for submitting your manuscript to PLOS ONE. After careful consideration, we feel that it has merit but does not fully meet PLOS ONE’s publication criteria as it currently stands. Therefore, we invite you to submit a revised version of the manuscript that addresses the points raised during the review process.

We look forward to receiving your revised manuscript.

Kind regards,

Alessio Paffoni, PhD

Academic Editor

PLOS ONE

Journal Requirements:

 “SHH received funding from Kementrian Riset, Teknologi dan Pendidikan Tinggi

Republik Indonesia (ID) (2018) and Universitas Gadjah Mada (2020).”

4. Please ensure that you refer to Figure 1 in your text as, if accepted, production will need this reference to link the reader to the figure.

Additional Editor Comments:

In order to address the review in the best possible way, the following aspects should be clarified:

- how was the sample size calculated?

- define the "baseline" time for measuring the basal levels of vitamin D

- provide the coefficients of variability for the measurement of serum 25-OH-D

- avoid repeating the same results in the text and tables

- indicate in the tables (example table 2) if statistical tests have been performed between different groups and if the test is significant (eg vitamin D among nullipara, primipara and multipara).

- analyze in more detail the differences between baseline and post-chemothrapy using median value and interquartile range. Also statistically define the difference.

- consider the time elapsed between baseline and final dosing and assess whether the seasonality of the sampling may have influenced the result.
---

## [Author Response · Author response to Decision Letter 0]

27 Mar 2022

Dear Editor,

We are thankful for the positive feedback received from the editorial team, and for the opportunity to respond to the constructive points in our submitted manuscript. Please find below our point-by-point response to the feedback outlining, where relevant, the related changes we have made. We have uploaded revised versions of the manuscript as instructed, including both a clean copy and a track changes version, and additional supplementary files made accordingly.

Comments

Journal requirements 

Response

Thank you for your remark. We have checked these requirements prior to submitting the revised manuscript.

2. Thank you for stating the following financial disclosure: “SHH received funding from Kementrian Riset, Teknologi dan Pendidikan Tinggi Republik Indonesia (ID) (2018) and Universitas Gadjah Mada (2020).” Please state what role the funders took in the study. If the funders had no role, please state: "The funders had no role in study design, data collection and analysis, decision to publish, or preparation of the manuscript." If this statement is not correct you must amend it as needed. Please include this amended Role of Funder statement in your cover letter; we will change the online submission form on your behalf.

Response: We confirm that this study only received funding from the stated institutions and the funder indeed had no role in the study design, data collection and analysis, decision to publish, or preparation of the manuscript. We also included the statement in our cover letter. 

3. In your Data Availability statement, you have not specified where the minimal data set underlying the results described in your manuscript can be found. PLOS defines a study's minimal data set as the underlying data used to reach the conclusions drawn in the manuscript and any additional data required to replicate the reported study findings in their entirety. All PLOS journals require that the minimal data set be made fully available. For more information about our data policy, please see http://journals.plos.org/plosone/s/data-availability. Upon re-submitting your revised manuscript, please upload your study’s minimal underlying data set as either Supporting Information files or to a stable, public repository and include the relevant URLs, DOIs, or accession numbers within your revised cover letter. For a list of acceptable repositories, please see http://journals.plos.org/plosone/s/data-availability#loc-recommended-repositories. Any potentially identifying patient information must be fully anonymized. Important: If there are ethical or legal restrictions to sharing your data publicly, please explain these restrictions in detail. Please see our guidelines for more information on what we consider unacceptable restrictions to publicly sharing data: http://journals.plos.org/plosone/s/data-availability#loc-unacceptable-data-access-restrictions. Note that it is not acceptable for the authors to be the sole named individuals responsible for ensuring data access. We will update your Data Availability statement to reflect the information you provide in your cover letter.

Response:

We have provided the minimal data set that were used to generate all tables and supplementary tables demonstrated the findings presented in the paper. The full de-identified dataset would not be able to be shared publicly due to restrictions imposed by the ethics committee as most of these are contain patient data, albeit de-identified, and it may be possible to determine the identify of participants given the extent of sociodemographic and clinical data available for each participant. 

Should there be a request for data, this can be sent to the corresponding author (email: susanna.hutajulu@ugm.ac.id). Future researchers can contact the institutional ethics committee (email: mhrec_fmugm@ugm.ac.id) at Universitas Gadjah Mada, Indonesia, with data access queries as well. 

In the revised manuscript we have uploaded the minimal data set as Supporting Information file. We also restated our data availability statement in the cover letter.

4. Please ensure that you refer to Figure 1 in your text as, if accepted, production will need this reference to link the reader to the figure.

Response: Thank you for your suggestion, the flow of subjects’ enrolment in the study as presented in Figure 1 is now cited in line 196.

Additional Editor comments

In order to address the review in the best possible way, the following aspects should be clarified

1. How was the sample size calculated?

Response: 

Thank you for your question about the sample size calculation used in the study. The sample calculation was performed to assess the main outcome in the study, observing changes in post-treatment vitamin D level. The minimal sample size was calculated to compare two paired means using paired t-test analysis, using standard deviation of differences obtained from previous study by Gabr and Marei (2017). We calculated the minimal sample to present 90% power with 5% significance. Although the minimal sample size required is 119 subjects, all subjects meeting the inclusion criteria from the main study is included to increase the power of data presented, making the final number of samples of the study into 136 subjects. We have now added in lines 164-166 under Methods.

2. Define the "baseline" time for measuring the basal levels of vitamin D.

Response: 

In this study, baseline blood sampling was taken within the time of subjects’ diagnosis until prior to the first dose of chemotherapy administration. The proposed timeline was within one week. However, during COVID 19 pandemic this timeline was altered. In some cases chemotherapy initiation was postpone while baseline blood sampling has already been performed, resulting in longer time space between sampling and chemotherapy administration. Same problem occured for post chemotherapy blood sampling when patients did not visit hospital in their planned schedule. Sampling timeline ranged 0-147 days before first chemotherapy and 3-125 days after the last chemotherapy. We now have made the definition of the baseline observation in the study clearer. It is now added in lines 144-150 under Methods.

3. Provide the coefficients of variability for the measurement of serum 25-OH-D.

Response: 

Thank you for your query on the coefficients of variability in the study. The intra- and inter-assay coefficient of variability (%CV) was 4.7% and 10.2%, respectively. This information is now disclosed in lines 155-156 under Methods.

4. Avoid repeating the same results in the text and table.

Response:

Thank you for this comment. We have reviewed our manuscript and made necessary adjustments to avoid repetition of the results already presented in all tables and modified Fig 1 legend.

5. Indicate in the tables (example table 2) if statistical tests have been performed between different groups and if the test is significant (eg vitamin D among nullipara, primipara and multipara).

Response:

Thank you for your suggestion. In this study, we compared the mean vitamin D between groups using independent t-test and ANOVA test and found no significant difference on all variables observed. We have added this information accordingly under Methods in lines 169-172 and legends of Table 2 and Table 4. We have also added the result of between groups comparison of baseline vitamin D under Results in lines 236-238, citing the corresponding results presented as S1 Table, and comparison of post-chemotherapy vitamin D level in lines 288-290, citing S5 Table.

6. Analyze in more detail the differences between baseline and post-chemotherapy using median value and interquartile range. Also statistically define the difference.

Response:

Thank you for your advice to convey the differences of vitamin D level in more details. Under Results in lines 255-258, we explained that the paired t-test analysis found a significant change of the vitamin D concentration, as compared to the level measured at baseline (p <0.001). The median and interquartile value obtained from computing the absolute changes of vitamin D between the two observation points was 3.13±4.03 ng/ml.

7. Consider the time elapsed between baseline and final dosing and assess whether the seasonality of the sampling may have influenced the result.

Response:

We have performed additional analysis to evaluate the time elapsed between sample collection and the closest chemotherapy completion, as well as the seasonality of the samples in our study. The time elapsed was calculated as the days between the sample collection and the closest last chemotherapy administration to the subjects. For the seasonality analysis, we matched the date of the sample collection with the season as determined by the national Meteorological, Climatological, and Geophysical Agency for the area of Yogyakarta, where the study took place. The analysis was performed using independent t-test as described under Methods in lines 171-175 and presented in S3 and S4 Tables as cited under Results in lines 265-271. 

We hope that, following our revisions, the manuscript is now suitable for publication in the PLOS ONE. 

Yours sincerely,

Susanna Hutajulu, MD, PhD

---

## [Decision Letter · Decision Letter 1]

17 May 2022

PONE-D-21-36153R1Vitamin D status in breast cancer cases following chemotherapy: a pre and post observational study in a tertiary hospital in Yogyakarta, IndonesiaPLOS ONE

Dear Dr. Hutajulu,

Thank you for submitting your manuscript to PLOS ONE. After careful consideration, we feel that it has merit but does not fully meet PLOS ONE’s publication criteria as it currently stands. Therefore, we invite you to submit a revised version of the manuscript that addresses the points raised during the review process.

ACADEMIC EDITOR:I agree with the reviewer that some important points deserve clarification.==============================

We look forward to receiving your revised manuscript.

Kind regards,

Alessio Paffoni, PhD

Academic Editor

PLOS ONE

Reviewers' comments:

Reviewer's Responses to Questions

**Comments to the Author**

1. If the authors have adequately addressed your comments raised in a previous round of review and you feel that this manuscript is now acceptable for publication, you may indicate that here to bypass the “Comments to the Author” section, enter your conflict of interest statement in the “Confidential to Editor” section, and submit your "Accept" recommendation.

Reviewer #1: (No Response)

2. Is the manuscript technically sound, and do the data support the conclusions?

Reviewer #1: Partly

3. Has the statistical analysis been performed appropriately and rigorously? 

Reviewer #1: Yes

4. Have the authors made all data underlying the findings in their manuscript fully available?

Reviewer #1: Yes

5. Is the manuscript presented in an intelligible fashion and written in standard English?

Reviewer #1: Yes

6. Review Comments to the Author

Reviewer #1: Manuscript ID PONE-D-21-36153

Title: Vitamin D status in breast cancer patients following chemotherapy: a pre and post

observational study in a tertiary hospital in Yogyakarta, Indonesia

In this study, the authors investigated To observe pre- and post-treatment vitamin D level and factors that influence its level changes after chemotherapy in breast cancer patients.

Comments for the Author:

1- The title is OK

2- In abstract: the objective differs from that of the text.

3- In Introduction is ok.

4- In methods:

Has the main study been published?

Better describe the inclusion and exclusion criteria in the present study.

Describe the sample size calculation

What is the time between the first and second serum vitamin D dosage?

5- In Results:

Age factor is important in terms of serum vitamin D values. What is the age range among the participants?

How many women were under 40 years of age?

How long is post-menopause?

What is the impact of weight on serum vitamin D values? Because 30% of women were obese.

In table 2, the presentation of the results is not clear.

In tables 2, 3 and 4, add statistical analysis with p value

6- In Discussion:

Could the type of test (ELISA) that was used for serum levels of vitamin D have influenced such low vitamin D results?

7. PLOS authors have the option to publish the peer review history of their article (what does this mean?). If published, this will include your full peer review and any attached files.

Reviewer #1: No

---

## [Author Response · Author response to Decision Letter 1]

26 May 2022

Dear Editor and Reviewer,

We are thankful for the positive feedback received from the reviewer, and for the opportunity to respond to the constructive points in our submitted manuscript. Please find below our point-by-point response to the feedback outlining, where relevant, the related changes we have made. We have uploaded revised versions of the manuscript as instructed, including both a clean copy and a track changes version, and additional supplementary files made accordingly.

Reviewer #1

1. The title is OK

Response

Thank you for your feedback.

2. In abstract: the objective differs from that of the text.

Response: 

The objective in the Abstract previously stated as “to observe pre- and post-treatment vitamin D level and factors that influence its changes after chemotherapy in breast cancer patients”, is now changed into “to observe pre -and post-treatment vitamin D level and its association with treatment and concomitant factors in breast cancer patients treated with chemotherapy” (lines 28-30)”

3. In introduction is ok.

Response: 

Thank you for reviewing this section.

4. In methods: Has the main study been published?

Response:

Thank you for your query on the main study. The prospective cohort is still ongoing, thus main data are still under collection. Nested studies exploring diagnostic delay in breast cancer (Hutajulu SH et al., PLoS One 2021) and patients’ experience (Prabandari et al., The Breast 2022) have been published as part of the main study.

5. In methods: Better describe the inclusion and exclusion criteria in the present study.

Response:

Thank you for your advice to elucidate the inclusion and exclusion criteria employed in the study. We try to make clear that the subjects included in our study are selected from an ongoing cohort with already selected criteria. Under Methods in lines 98-109, we now explain that “We performed a nested pre-post observational study assessing 25(OH)D changes on primary breast cancer patients registered in a prospective ongoing cohort. The main study analysed the risk of chemotherapy side effects and its effect on survival and quality of life in breast cancer patients. Subjects of the main study were histologically confirmed female breast cancer patients aged ≥18 years who were chemotherapy naïve and receiving their first chemotherapy in the Haematology and Medical Oncology Division, “Tulip”/Integrated Cancer Clinic, Dr. Sardjito General Hospital, Yogyakarta, Indonesia, from 2018-2022. No subjects in the cohort were in a terminal condition or with severe congestive heart failure. Among subjects enrolled in the main cohort, we only included those who have received their last chemotherapy cycle and completed post-treatment follow-up into the present study. Subjects who have incomplete blood sample were excluded.”

6. In methods: Describe the sample size calculation

Response:

Thank you for this suggestion. We performed the calculation of sample size to achieve the minimum number of samples for the main outcome. The minimal sample size was calculated for paired mean comparison using paired t-test analysis. The standard deviation of differences was obtained from a previous study by Gabr and Marei (2017), who found the mean difference between pre- and post-treatment vitamin D in Egyptian breast cancer patients to be 5.51 ng/ml (SD=6.67 ng/ml). We calculated the minimal sample to present 90% power with 5% significance. Although the minimal sample size required is 119 subjects, all subjects meeting the inclusion criteria from the main study are included to increase the power of data presented, making the final number of samples of the study 136 subjects. The sample size calculation is briefly explained under Methods in lines 170-172. We have also added the referenced study as a reference [26].

7. In methods: What is the time between the first and second serum vitamin D dosage?

Response:

Thank you for your question. We would like to clarify that no vitamin D supplementation was provided to the subjects as part of our study. However, we do perform the vitamin D measurement in two observation points, prior to- and after chemotherapy. The median duration between pre- and post-treatment sample collection in our study was 5.00±1.64 months. Although vitamin D changes between observation points are lower on samples taken more than 5 months apart (median±IR =2.46±3.31 ng/ml) as compared to less than 5 months (median±IR =3.59±4.12 ng/ml), we found that the difference is not statistically significant (p =0.055). While no significant difference in vitamin D concentration was found based on the duration of the delay in collecting blood samples (S3 Table), this delay prolongs the distance between pre- and post-treatment observation and might not represent the actual duration of the chemotherapy received by subjects. Instead, we report our findings by stratifying the vitamin D changes based on the number of chemotherapy cycles administered to better represent the neurotoxicity presumed to cause the decrease of vitamin D concentration in subjects.

8. In results: Age factor is important in terms of serum vitamin D values. What is the age range among the participants? How many women were under 40 years of age?

Response:

We agree that age is very important in regards to vitamin D levels. In our study, the median subjects’ age is 51±13 years old. The range of the age is 46, the youngest participant was 32 and the oldest was 78 years old. Based on the Spearman analysis result as presented in Table 2, age did not correlate with the baseline vitamin D level in our study (r =0.096, p =0.266). After stratifying based on the median age, the vitamin D level remains similar among younger and older subjects, both at baseline (p =0.428) and post-treatment (p =0.755). In our study, only 12 (8.82%) subjects were under 40 years old. Vitamin D however remains similar among groups under 40 and older (p =0.753). This result is added under Results in lines 248-249.

9. In results: How long is post-menopause?

Response:

Although 72 subjects were already menopause at the beginning of the study, only 67 (93.1%) were able to recall or have any documented examination results of their menopause. Among them, the median age of menopause is 50±6 years old. The median duration of menopause is 7.91±10.5 years when baseline vitamin D was measured. This result is now added under Results in lines 210-213. 

10. In results: What is the impact of weight on serum vitamin D values? Because 30% of women were obese.

Response:

In our study, the low vitamin D level was observed in all subjects, and no difference was found among BMI categories. The median vitamin D did not differ (p =0.819) among subjects who were underweight (median±IR =8.06±5.44 ng/ml), normal (median±IR =9.10±5.27 ng/ml), overweight (median±IR =9.68±3.16 ng/ml), and obese (median±IR =8.23±2.82 ng/ml). However, we did not observe the body fat distribution of the subjects in our study. The higher proportion of visceral adipose tissue despite lower BMI in the Asian population and its stronger correlation with vitamin D instead of the overall body fat might explain the low vitamin D across different BMI as found in our study. This is now added under Results in lines 250-253 and Discussion in lines 369-372.

11. In results: In table 2, the presentation of the results is not clear.

Response:

Thank you for your remarks. In Table 2, we try to present the result of the Spearman correlation analysis result among the socio-demographic and clinicopathological characteristics of the subjects with their baseline vitamin D concentration. The value of baseline vitamin D, as well as the variable age, age menarche, age of first pregnancy, upper arm circumference, Hb, albumin, leucocyte, AST, and ALT, were analysed as continuous data. The variable menopause, parity, education, occupation, insurance, BMI, histological type, grade, tumor size, stage, ER, PR, and HER2 were analysed as categorical data, with the median vitamin D concentration of each category also presented in the table. The result is presented in Spearman rho and p-value. Among the factors observed, only AST shows a significant correlation. In addition to this analysis, we also compare the median vitamin D levels among different categories in each group using an independent t-test and ANOVA test. This result is presented in S1 Tables.

12. In results: In tables 2, 3 and 4, add statistical analysis with p value

Response:

Thank you for your advice. We now have added the statistical analysis corresponding to each p-value reported in Tables 2, 3, and 4.

13. In discussion: Could the type of test (ELISA) that was used for serum levels of vitamin D have influenced such low vitamin D results?

Response:

Thank you for expressing this concern. We choose the ELISA method to perform vitamin D analysis in our study due to its high availability in our country and to increase its reproducibility and the feasibility to apply the methods directly to clinical service in our setting. The kit we selected has high specificity (74.7% for 25-OH-D2 and 100% for 25-OH-D3) and good agreement with Diasorin LIAISON chemiluminescent immunoassay (CLIA) (R =0.919) and Roche Cobas (ECLIA) (R =0.948) in measuring vitamin D concentration. The respective assays have high correlations with liquid chromatography-tandem mass spectrometry (LC-MS/MS) (R =0.9455 and R =0.9102), which currently has the highest sensitivity and specificity in vitamin D measurement.

In performing the assay, we used low and high control in duplicates on each plate. The range of low and high control provided by the kit was 7.28-15.1 ng/ml and 37.9-78.7 ng/ml, respectively. In every plate, the low (ranged 7.02-11.57 ng/ml) and high (ranged 50.45-55.61 ng/ml) controls measured fell within the QC ranges. In addition, the assay was performed with 0-120 ng/ml standards which were provided in 6 concentrations; 0, 5, 15, 30, 60, and 120 ng/ml. In performing the assay, the lowest sample measured was 0.68 ng/ml and the highest sample measured was 25.16 ng/ml. 

All samples were kept in -80 prior to analysis with no history of freeze-thaw. Furthermore, our plasma samples, QC low, QC high, and standards from the kits were treated uniformly and concurrently, minimising any possible treatment biases of the assay. Therefore, the vitamin D concentration of the samples in this study falls within the range of the standard curve and we are confident that the kit could not possibly contribute to the low values of the samples measured.

Considering the similarly low vitamin D concentration observed in Korean breast cancer patients in a study by Kim et al., in 2018 (median =12.94 ng/ml) and the high prevalence of deficiency in various studies in different at-risk Indonesian, the very low vitamin D concentration observed in our study suggest the severity of low vitamin D in Indonesian women. This information about the assay is now added under Methods in lines 170-172 and Discussion in lines 383-393.

Additional remarks

In the previous manuscript version, we mislabelled the stratified groups based on subjects’ menopausal status in Table 1. Necessary adjustments have been made in lines 4-5 of Table 1 and under Results in line 207. We confirm that this error does not affect other results presented in the manuscript.

We hope that, following our revisions, the manuscript is now suitable for publication in the PLOS ONE. 

Yours sincerely,

Susanna Hutajulu, MD, PhD

---

## [Decision Letter · Decision Letter 2]

12 Jun 2022

Vitamin D status in breast cancer cases following chemotherapy: a pre and post observational study in a tertiary hospital in Yogyakarta, Indonesia

PONE-D-21-36153R2

Dear Dr. Hutajulu,

We’re pleased to inform you that your manuscript has been judged scientifically suitable for publication and will be formally accepted for publication once it meets all outstanding technical requirements.

Kind regards,

Alessio Paffoni, PhD

Academic Editor

PLOS ONE

Additional Editor Comments (optional):

Reviewers' comments:

Reviewer's Responses to Questions

**Comments to the Author**

1. If the authors have adequately addressed your comments raised in a previous round of review and you feel that this manuscript is now acceptable for publication, you may indicate that here to bypass the “Comments to the Author” section, enter your conflict of interest statement in the “Confidential to Editor” section, and submit your "Accept" recommendation.

Reviewer #1: All comments have been addressed

2. Is the manuscript technically sound, and do the data support the conclusions?

Reviewer #1: Yes

3. Has the statistical analysis been performed appropriately and rigorously? 

Reviewer #1: N/A

4. Have the authors made all data underlying the findings in their manuscript fully available?

Reviewer #1: Yes

5. Is the manuscript presented in an intelligible fashion and written in standard English?

Reviewer #1: Yes

6. Review Comments to the Author

Reviewer #1: The authors answered all questions. They made the requested changes and corrections. The paper presented was reviewed by the authors and has all corrections

7. PLOS authors have the option to publish the peer review history of their article (what does this mean?). If published, this will include your full peer review and any attached files.

Reviewer #1: No

---

## [Editor Report · Acceptance letter]

15 Jun 2022

PONE-D-21-36153R2 

Vitamin D status in breast cancer cases following chemotherapy: a pre and post observational study in a tertiary hospital in Yogyakarta, Indonesia 

Dear Dr. Hutajulu:

I'm pleased to inform you that your manuscript has been deemed suitable for publication in PLOS ONE. Congratulations! Your manuscript is now with our production department. 

Kind regards, 

on behalf of

Dr. Alessio Paffoni 

Academic Editor

PLOS ONE